# Factors associated with in-hospital mortality of adult tetanus patients–a multicenter study from Bangladesh

Md. Abdullah Saeed Khan[1,2]*, Mohammad Jahid Hasan[2], Md. Utba Rashid[3], Soumik Kha Sagar[3], Sanzida Khan[1], Susmita Zaman[2], Sultan Mahamud Sumon[1], Ariful Basher[1], Mohammad Delwer Hossain Hawlader[4], Mohammad Hayatun Nabi[4], Nadira Sultana Kakoly[4]

1 Infectious Disease Hospital, Dhaka, Bangladesh, 2 Pi Research Consultancy Center, Dhaka, Bangladesh, 3 Nutrition and Clinical Services Division (NCSD), International Centre for Diarrhoeal Disease Research, Bangladesh (icddr,b), Dhaka, Bangladesh, 4 Department of Public Health, North South University, Dhaka, Bangladesh

* abdullahdmc@gmail.com

**Data Availability Statement:** All relevant data are within the manuscript and its Supporting Information files.

## Abstract

### Background

Tetanus, a vaccine-preventable disease, is still occurring in the elderly population of low- and middle-income countries with a high case-fatality rate. The objective of the study was to elucidate the factors associated with in-hospital mortality of tetanus in Bangladesh.

### Methods

This prospective observational study, conducted in two specialized infectious disease hospitals, conveniently selected adult tetanus patients ($\geq$18 years) for inclusion. Data were collected through a preformed structured questionnaire. Kaplan Meier survival analysis and univariate and multivariable Cox regression analysis were carried out to assess factors associated with in-hospital mortality among patients. All analysis was done using Stata (version 16) and SPSS (version 26).

### Results

A total of 61 tetanus cases were included, and the overall in-hospital mortality rate was 34.4% (n = 21). Patients had an average age of 46.49 ±15.65 years (SD), and the majority were male (96.7%), farmers (57.4%), and came from rural areas (93.4%). Survival analysis revealed that the probability of death was significantly higher among patients having an age of $\geq$ 40 years, incubation time of $\leq$12 days, onset time of $\leq$ 4 days, and having complication(s). However, on multivariable Cox regression analysis, age (adjusted hazard ratio [aHR] 4.03, 95% Confidence Interval [CI] 1.07–15.17, p = 0.039) and onset time ($\leq$4 days) (aHR 3.33; 95% CI 1.05–10.57, p = 0.041) came as significant predictors of in-hospital mortality after adjusting for incubation period and complications.

**Funding:** The author(s) received no specific funding for this work.

**Competing interests:** The authors have declared that no competing interests exist.

## Conclusion

Older age and short onset time are the two most important determinants of in-hospital mortality of tetanus patients. Hence, these patients require enhanced emphasis and care.

## Author summary

The number of new cases of tetanus occurring every year decreased considerably in many countries through mass vaccination programs. However, it continues to cause deaths of many people in low- and middle-income settings. The majority of hospital-admitted adult (≥18 years) tetanus patients included in this study were male, farmer and came from rural areas. More than one-third of them died within hospital. However, one must note that these hospitals lacked intensive care facilities. Tetanus mortality after hospitalization is dependent on many factors. Our study found that patients' older age, short incubation period (time from injury to the appearance of symptom), short onset time (interval between the first symptoms and the first spasm), and development of complications were significant contributors of in-hospital deaths. Meticulous and individualized management of adult tetanus patients with one or more of the above features is required to increase their survival. Moreover, adult males from high risk occupation could be potential targets for booster vaccination strategies to prevent incidence of tetanus in Bangladesh.

## Introduction

Tetanus, an acute and fatal infection, is caused by the neurotoxin-producing bacterium *Clostridium tetani*. It shows a classical clinical picture of muscular rigidity and generalized spasms [1]. The spores of *C. tetani* are resilient, long-lasting, and widespread in the environment. It can contaminate wounds, abrasions, and the umbilical stump in neonates [2]. Globally approximately one hundred thousand people contracted tetanus in 2017 [3]. It has a high case fatality rate, with an estimated 45.5% (95% CI: 43.7%-47.2%) deaths in African countries [4]. However, it might vary based on the availability and accessibility of well-equipped intensive care units. Extensive vaccination coverage has led to a decline in the number of new cases of tetanus in developed countries. However, it is still common in low- and middle-income countries. South Asia and Sub-Saharan Africa account for 82% of tetanus cases worldwide [3]. In Bangladesh the prevalence is still unknown [5].

Any unvaccinated person has the potential risk of developing tetanus because of the absence of immune protection from natural infection. As childhood and maternal vaccination, started at the end of the twentieth century, has rapidly reduced the incidence of tetanus throughout the world, it continues to be a substantial problem among older adults in many countries [2]. Moreover, neonatal tetanus, which occurs primarily due to inadequate or lack of women's immunization, is frequently found in developing or underdeveloped settings [6]. Despite the availability of inexpensive and effective tetanus vaccines, the disease continues to be a health problem in impoverished regions of the world [7].

*C tetani* spores invade the human body through wounds or minor abrasions under suitable anaerobic conditions. Whenever a wound occurs, appropriate wound care and vaccination could prevent tetanus [8]. If infection occurs, patients often die in the hospital due to autonomic failure, cardiovascular dysrhythmia, and complications [9]. Proper management can

improve survival among patients [10]. However, despite appropriate management, many patients fail to survive.

Death in tetanus is dependent on many factors. The poor prognostic factors of in-hospital deaths identified are short incubation period, older age, severe type, generalized variety, dysautonomia, pneumonia, hypoxemia, sepsis, and renal failure [11]. Hence, the treatment of cases often requires assisted ventilation. Additionally, passive immunization is usually given following trauma or injury [12]. Despite widespread vaccination and advances in management, the mortality in generalized tetanus is still high. Previously, very few studies prospectively explored the factors associated with in-hospital mortality among adult tetanus patients. Moreover, there are few to no studies regarding in-hospital mortality and its associated factors among adult tetanus patients in Bangladesh. Therefore, this research aimed to analyze the factors affecting mortality in hospitalized adult tetanus patients.

## Materials and methods

### Ethics statement

The study was approved by the Ethical Review Committee of North South University (2020/OR-NSU/IRB-No.0801). All procedures were conducted following guidelines laid out by the World Medical Association Declaration of Helsinki. Informed written consent was obtained from patients' caregivers if the patient was unconscious or unable to write. However, in the latter case, a verbal consent was taken from the patient first.

### Study design, population, and settings

This prospective observational was carried out in two specialized infectious disease hospitals in Bangladesh (Infectious Disease Hospital, Mohakhali, Dhaka, and Surya Kanto Hospital, Mymensingh), between December 2020 to August 2021. Both of these hospitals have dedicated units (wards) for tetanus patients, including all management facilities except the provision of intensive care and assisted ventilation. We approached all adult (≥18 years) hospitalized cases of tetanus for inclusion. Patients who were not willing to participate were excluded. A total of 78 patients were admitted during the study period. However, 16 patients didn't match the age criteria (i.e., they were <18 years) and one patient didn't give consent. Finally, a total of 61 clinically diagnosed tetanus patients were conveniently selected for the study.

### Study measures

After an extensive review of the published literature, we produced a structured questionnaire for data collection (**S1 Table**). The questionnaire had four parts: a.) sociodemographic information, b.) information regarding tetanus, c.) comorbidities, investigations, and treatment, d.) in-hospital outcome.

### Sociodemographic information

This part queried the patient's age, sex, religion, education, marital status, occupation, monthly family income, and smoking history.

### Information regarding tetanus

This part asked about the mode of injury (trauma or surgery), site of the wound, clinical features (including symptoms and signs) at admission, vaccination history (both previous and post-exposure), and investigations. The clinical characteristics comprised of tetanus types (localized or generalized), the incubation period (time from injury to the appearance of

symptom), onset time (interval between the first symptoms and the first spasm), trismus (lockjaw), risus sardonicus (a characteristic, abnormal, sustained spasm of the facial muscles), dysphagia, muscle spasms (other skeletal muscles), spasticity, rigidity (overall), abdominal rigidity, opsithotonus (spasm of the muscles causing backward arching of the head, neck, and spine), fever, palpitation, urinary retention and vital signs. Severity of tetanus was assessed at admission using Ablett's classification of tetanus severity [2] (**S2 Table**). Severity of trismus was defined as follows- mild trismus (minimum mouth opening: 30–40 nm or 2–3 fingers), moderate trismus (15–30 nm or 1–2 fingers), and severe trismus (<15 mm or <1 finger). Localized tetanus was defined as spasms localized to extremities or the head, and generalized tetanus was defined as generalized muscular spasms all over the body [2]. History of tetanus vaccination and/or immunoglobulin administration after injury was termed as post-exposure prophylaxis.

## Comorbidities, investigations, and treatment

Patients' comorbidities, including diabetes mellitus, hypertension, chronic obstructive pulmonary disease, chronic renal failure, stroke, and ischemic heart disease, were assessed in this section. Additionally, the information about routine investigations (hematological profile, serum creatinine, serum calcium, and electrolytes) and treatment received during the hospital stay was also collected.

## In-hospital outcome

Complications and final outcomes of the patients were listed in this section.

## Study procedure

The data collection was done by the principal investigator (PI) (MASK) and three co-investigators (UBR, SKS, and SK) of the research. All of them are physicians directly involved in the management and research of tetanus patients in the study centers. All data collectors were trained on the questionnaire before the study started to maintain uniformity of data collection. Moreover, the PI randomly checked the collected data forms to check for errors at regular intervals. After taking informed written consent we recorded patients' socio-economic data, vaccination history, clinical characteristics, and comorbidity data upon inclusion. Patients presenting with lockjaw (trismus) or risus sardonicus and one or more features from dysphagia, muscle spasms, abdominal rigidity, opisthotonus, and history of the wound were considered to have clinically confirmed tetanus. The presence of an infected wound was considered an essential diagnostic clue in the absence of trismus or risus sardonicus. After the initial assessment, patients were followed up during the hospital stay until discharge, with recovery or death as the final outcome. Finally, investigation reports and in-hospital outcome data, including complications, were recorded in the questionnaire.

## Statistical analysis

After data entry and curation, we performed descriptive and analytic statistics. Descriptive statistic was expressed as frequency (proportion) for categorical variable and mean ± standard deviation or median (interquartile range) for continuous variable. Univariate analyses were conducted using the chi-square test, independent samples t test, Mann-Whitney U test, and Kaplan Meier Survival analysis. Univariate and multivariable Cox regression analyses were used for the assessment of significant factors associated with death. Only statistically significant factors ($p < 0.05$) at univariate analysis, including age, incubation period, onset time, and

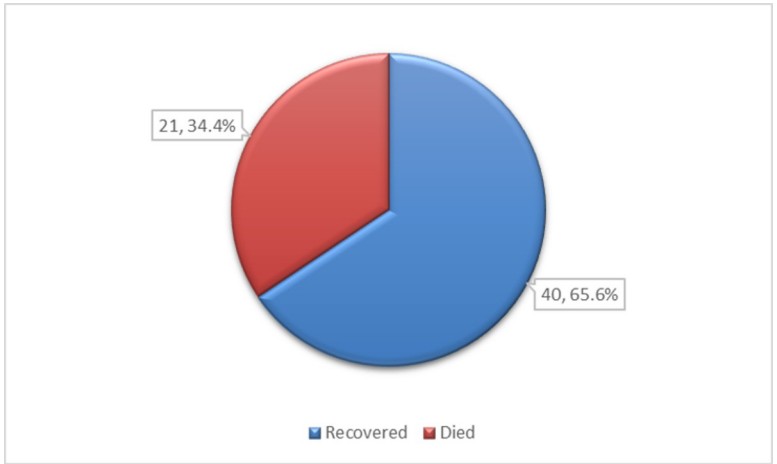

**Fig 1. In-hospital mortality of tetanus patients (n = 61).**

presence of complication, were considered for Cox regression. A p-value of ≤0.05 was considered significant. Statistical analysis was performed using statistical software Stata (version 16).

## Result

The prevalence of in-hospital mortality was 34.4% (21 out of 61) in this study. (**Fig 1**).

The average age of all patients was 46.49±15.65 years. The mean age of patients who died (52.10 ±12.99 years) was significantly higher than that of recovered patients (43.55 ±16.26 years, p = 0.042). Overall, 63.9% were aged ≥40 years. However, among patients who died, 80.9% were aged ≥ 40 years, and among those who were alive, 55.0% had the same age (p = 0.045). The majority patients were male (96.7%), illiterate (46.7%), married (76.7%), farmer (57.4%), had income <7500 BDT (US$ <87) (42.6%), and came from rural area (93.4%). The distribution was statistically similar among patients who were alive and who died (**Table 1**).

Of all patients, 6.7% had hypertension, 4.9% had diabetes mellitus, 1.6% had COPD, and 1.6% had ischemic heart disease. Total 8.2% of patients had at least one comorbidity. Thirty-six percent of patients were current smokers, 20% were past smokers, and 44% never smoked (**Table 2**).

**Table 3** presents the clinical characteristics of the patients. Only 6.6% of patients asserted a history of vaccination against tetanus, and 24.6% of patients took vaccine after an injury (post-exposure prophylaxis). The main mode of injury was trauma (95.1%), and the rest had surgeries. The majority had their wound in the extremities (91.8%). Most of the patients had the generalized type of tetanus (93.4%) and severe disease (45.9%). These features were statistically similar between alive and dead patients. The overall median onset time was 3 days (interquartile range [IQR]: 2–7). It was statistically similar among patients who died (2.5, IQR 2–4) than those who were alive (4, IQR 2–10, p = 0.076). The median incubation period was 14 days (IQR 8–15) overall. It was also statistically similar between deceased patients (12, IQR 7–15) and alive patients (15, IQR 10–16, p = 0.058). However, when categorized at a cutoff point of 12 days, a statistically significantly higher proportion of dead patients had an incubation period of ≤12 days compared to those alive (p = 0.043). Of all patients, the majority had severe trismus (33.9%), mild dysphagia (50%), short spasms (73.5%), and generalized rigidity (93.4%). Spasticity was present in 75.4% of patients, abdominal rigidity in 68.9%, fever in 41%, opisthotonus in 9.8%, and urinary retention in 9.8%. Vital signs were within the normal range.

**Table 1. Sociodemographic characteristics of the patients (n = 61).**

| Variable | Total n (%) | Recovered n (%) | Died n (%) | p-value |
|---|---|---|---|---|
| **Age (years), mean±SD** | 46.49 ±15.65 | 43.55 ±16.26 | 52.10 ±12.99 | **0.042** |
| **Age category (years)** | | | | |
| < 40 | 22 (36.1) | 18 (45.0) | 4 (19.1) | **0.045** |
| ≥ 40 | 39 (63.9) | 22 (55.0) | 17 (80.9) | |
| **Sex** | | | | |
| Male | 59 (96.7) | 38 (95.0) | 21 (100.0) | 0.297 |
| Female | 2 (3.3) | 2 (5.0) | 0 | |
| **Education** | | | | |
| Illiterate | 28 (46.7) | 18 (45.0) | 10 (50.0) | 0.986 |
| Primary | 16 (26.7) | 11 (27.5) | 5 (25.0) | |
| SSC | 13 (21.6) | 9 (22.5) | 4 (20.0) | |
| HSC and above | 3 (5.0) | 2 (5.0) | 1 (5.0) | |
| **Marital Status** | | | | |
| Unmarried | 9 (15.0) | 6 (15.4) | 3 (14.3) | 0.967 |
| Married | 46 (76.7) | 30 (76.9) | 16 (76.2) | |
| Widow/divorced | 5 (8.3) | 3 (7.7) | 2 (9.5) | |
| **Occupation** | | | | |
| Farmer | 35 (57.4) | 20 (50.0) | 15 (71.4) | 0.273 |
| Businessman | 6 (9.8) | 6 (15.0) | 0 | |
| Service holder | 5 (8.2) | 3 (7.5) | 2 (9.5) | |
| Carpenter | 4 (6.6) | 1 (2.5) | 3 (14.3) | |
| Housewife | 2 (3.3) | 2 (5.0) | 0 | |
| Student | 2 (3.3) | 1 (2.5) | 1 (4.8) | |
| Electrician | 2 (3.3) | 2 (5.0) | 0 | |
| Retired | 2 (3.3) | 2 (5.0) | 0 | |
| Tailor | 1 (1.6) | 1 (2.5) | 0 | |
| Driver | 1 (1.6) | 1 (2.5) | 0 | |
| Fisherman | 1 (1.6) | 1 (2.5) | 0 | |
| **Monthly family Income in BDT (US$)** | | | | |
| <7500 (<87$) | 26 (42.6) | 18 (45.0) | 8 (38.1) | 0.382 |
| 7501–10000 (87$-116$) | 13 (21.3) | 9 (22.5) | 4 (19.1) | |
| 10001–15000 (116$-175$) | 16 (26.2) | 11 (27.5) | 5 (23.8) | |
| >15000 (>175$) | 6 (9.8) | 2 (5.0) | 4 (19.0) | |
| **Residence** | | | | |
| Rural | 57 (93.4) | 39 (97.5) | 18 (85.7) | 0.077 |
| Urban | 4 (6.56) | 1 (2.5) | 3 (14.3) | |

p-value determined using independent samples t test and Chi-square test where appropriate; Significant p-values were shown in bold.

The distribution of the clinical features was statistically similar between patients who were alive and those who died.

Hematological profile, random blood sugar, serum creatinine, calcium, and electrolytes were statistically similar between alive and dead patients except for serum sodium which was significantly higher in patients who died (141.9±1.8 mmol/l) than that of those who were alive (138.0±0.9 mmol/l, p = 0.034). However, the average sodium level was within the normal range for both groups of patients (**Table 4**).

Out of 61 patients, 26 (42.6%) developed at least one complication, and this proportion was significantly higher in dead patients (61.9%) compared to that of alive patients (32.5%,

**Table 2. Comorbidity and smoking habit of the patients (n = 61).**

| Variable | Total n (%) | Recovered n (%) | Died n (%) | p-value |
|---|---|---|---|---|
| **Comorbidity** | | | | |
| Present (any) | 5 (8.2) | 2 (5.0) | 3 (14.3) | 0.209 |
| Absent | 56 (91.8) | 38 (95.0) | 18 (85.7) | |
| **Diabetes Mellitus** | | | | |
| Present | 3 (4.9) | 1 (2.5) | 2 (9.5) | 0.228 |
| Absent | 58 (95.1) | 39 (97.5) | 19 (90.5) | |
| **Hypertension** | | | | |
| Present | 4 (6.7) | 1 (2.5) | 3 (14.3) | 0.077 |
| Absent | 57 (93.4) | 39 (97.5) | 18 (85.7) | |
| **Chronic Obstructive Pulmonary Disease** | | | | |
| Present | 1 (1.6) | 0 | 1 (4.8)) | 0.164 |
| Absent | 60 (98.4) | 40 (100.0) | 20 (95.2) | |
| **Ischemic heart disease** | | | | |
| Yes | 1 (1.6) | 0 | 1 (4.8) | 0.164 |
| No | 60 (98.4) | 40 (100.0) | 20 (95.2) | |
| **Smoking habit*** | | | | |
| Current smoker | 9 (36.0) | 7 (36.8) | 2 (33.3) | 0.940 |
| Past smoker | 5 (20.0) | 4 (21.1) | 1 (16.7) | |
| Never smoker | 11 (44.0) | 8 (42.1) | 3 (50.0) | |

*After excluding missing cases

p-value determined by Chi-square test

p = 0.027). The most common complication was hypoxemia (31.1%) followed by aspiration pneumonia (19.6%), bedsore (13.1%), dysautonomia (6.5%), DVT (4.9%), UTI (3.28%), sepsis (1.6%), thrombophlebitis (1.6%), and wound infection (1.6%). The individual distribution of the complications was statistically similar between the alive and dead patient groups (**Table 5**).

The median duration of hospital stay of the patients was 11 days (IQR: 6–21 days) (**Fig 2**). Recovered patients had a significantly longer duration of stay (median 17 days; IQR: 10–23 days) than those who died (median 5 days; IQR: 2–8 days) (p<0.001).

Kaplan-Meier survival analysis showed that the probability of survival among tetanus patients at 40 days after admission was 0.6, and it was statistically significantly better among patients with an age of < 40 years, incubation time of >12 days, onset time of >4 days, and no complications (**Fig 3**).

Only factors that were significant (p<0.05) in Kaplan-Meier survival analysis were considered for univariate and multivariable Cox regression models to determine their associations with the in-hospital mortality of tetanus patients (**Table 6**). After adjusting for incubation period and complications, only age (≥40 years) and onset time (≤4 days) were found to be significant predictors of tetanus case-fatality in the hospital. Tetanus patients aged ≥40 years were 4.03 times (95% CI 1.07–15.17, p = 0.039) more likely to die due to tetanus than those aged <40 years. Patients with an onset time of ≤4 days were significantly more likely (aHR 3.33; 95%CI 1.05–10.57, p = 0.041) to die in the hospital than those with a higher onset time.

## Discussion

Tetanus is a vaccine-preventable disease that is still continuing to infect people, particularly, in low- and middle-income countries. As the spore of the causative organism *C. tetani* remains

**Table 3. Clinical characteristics of the patients (n = 61).**

| Variable | Total n (%) | Recovered n (%) | Died n (%) | p-value |
|---|---|---|---|---|
| **Prior history of vaccination against tetanus** | | | | |
| Present | 4 (6.6) | 3 (7.5) | 1 (4.8) | 0.566 |
| Absent | 19 (31.2) | 14 (35.0) | 5 (23.8) | |
| Don't know | 38 (62.3) | 23 (57.5) | 15 (71.4) | |
| **Postexposure prophylaxis** | | | | |
| Taken | 15 (24.6) | 10 (25.0) | 5 (23.8) | 0.602 |
| Not taken | 40 (65.6) | 25 (62.5) | 15 (71.4) | |
| Don't know | 6 (9.8) | 5 (12.5) | 1 (4.8) | |
| **Mode of injury** | | | | |
| Post-traumatic | 58 (95.1) | 37 (92.5) | 21 (100.0) | 0.209 |
| Post-surgery | 3 (4.9) | 3 (7.5) | 0 | |
| **Place of wound** | | | | |
| Extremities | 56 (91.8) | 36 (90.0) | 20 (95.2) | 0.479 |
| Head/face | 5 (8.2) | 4 (10.0) | 1 (4.8) | |
| **Tetanus type** | | | | |
| Generalized | 57 (93.4) | 37 (92.5) | 20 (95.2) | 0.681 |
| Localized | 4 (6.7) | 3 (7.5) | 1 (4.8) | |
| **Onset time (days), median (IQR)*** | 3 (2–7) | 4 (2–10) | 2.5 (2–4) | 0.076 |
| > 4 days | 20 (36.4) | 16 (45.7) | 4 (20.0) | 0.057 |
| ≤ 4 days | 35 (63.6) | 19 (54.3) | 16 (80.0) | |
| **Incubation period (days), median (IQR)*** | 14 (8–15) | 15 (10–16) | 12 (7–15) | 0.058 |
| > 12 days | 28 (52.8) | 21 (63.6) | 7 (35.0) | **0.043** |
| ≤ 12 days | 25 (47.2) | 12 (36.4) | 13 (65.0) | |
| **Tetanus severity** | | | | |
| Mild | 11 (18.0) | 8 (20.0) | 3 (14.3) | 0.871 |
| Moderate | 18 (29.5) | 12 (30.0) | 6 (28.6) | |
| Severe | 28 (45.9) | 18 (45.0) | 10 (47.6) | |
| Very severe | 4 (6.6) | 2 (5.0) | 2 (9.5) | |
| **Trismus*** | | | | |
| Absent | 1 (1.7) | 1 (2.6) | 0 | 0.677 |
| Mild | 8 (13.6) | 6 (15.4) | 2 (10.0) | |
| Moderate | 30 (50.8) | 21 (53.8) | 9 (45.0) | |
| Severe | 20 (33.9) | 11 (28.2) | 9 (45.0) | |
| **Risus sardonicus** | | | | |
| Present | 5 (8.2) | 4 (10.0) | 1 (4.8) | 0.479 |
| Absent | 56 (91.8) | 36 (90.0) | 20 (95.2) | |
| **Dysphagia*** | | | | |
| Absent | 6 (10.7) | 3 (8.6) | 3 (14.3) | 0.378 |
| Mild | 28 (50.0) | 20 (57.1) | 8 (38.1) | |
| Severe | 22 (39.3) | 12 (34.3) | 10 (47.6) | |
| **Spasms*** | | | | |
| Short | 25 (73.5) | 15 (71.4) | 10 (76.9) | 0.724 |
| Prolonged | 9 (26.5) | 6 (28.6) | 3 (23.1) | |
| **Spasticity** | | | | |
| Present | 46 (75.4) | 31 (77.5) | 15 (71.4) | 0.601 |
| Absent | 15 (24.6) | 9 (22.5) | 6 (28.6) | |
| **Rigidity** | | | | |

(*Continued*)

**Table 3.** (Continued)

| Variable | Total n (%) | Recovered n (%) | Died n (%) | p-value |
|---|---|---|---|---|
| Localized | 4 (6.6) | 3 (7.5) | 1 (4.8) | 0.681 |
| Generalized | 57 (93.4) | 37 (92.5) | 20 (95.2) | |
| **Abdominal rigidity** | | | | |
| Present | 42 (68.9) | 26 (65.0) | 16 (76.2) | 0.370 |
| Absent | 19 (31.2) | 14 (35.0) | 5 (23.8) | |
| **Opisthotonus** | | | | |
| Present | 6 (9.8) | 6 (15.0) | 0 | 0.062 |
| Absent | 55 (90.2) | 34 (85.0) | 21 (100.0) | |
| **Fever** | | | | |
| Present | 25 (41.0) | 15 (37.5) | 10 (47.6) | 0.445 |
| Absent | 36 (59.0) | 25 (62.5) | 11 (52.4) | |
| **Palpitation** | | | | |
| Present | 6 (9.8) | 4 (10.0) | 2 (9.5) | 0.953 |
| Absent | 55 (90.2) | 36 (90.0) | 19 (90.5) | |
| **Urinary retention** | | | | |
| Present | 6 (9.8) | 6 (15.0) | 0 | 0.069 |
| Absent | 55 (90.2) | 34 (85.0) | 21 (100.0) | |
| **Pulse (b/min),** mean±SD | 82.9 ±1.6 | 80.0 ±1.9 | 88.2 ±2.3 | **0.011** |
| **Systolic blood pressure (mmHg),** mean±SD | 111.4 ±1.9 | 109.0 ±2.1 | 115.8 ±3.6 | 0.087 |
| **Diastolic blood pressure (mmHg),** mean±SD | 73.0 ±1.4 | 73.9 ±1.9 | 71.5 ±1.9 | 0.433 |
| **Temperature (F),** mean±SD | 97.8 ±0.6 | 97.2 ±0.9 | 98.7 ±0.3 | 0.214 |
| **Respiratory rate** (breaths/min) | 20 ±0.6 | 19.5 ±0.8 | 20.9 ±0.9 | 0.256 |

*Excluding missing values

p-value determined by Mann-Whitney U test, independent samples t test and Chi-square test where appropriate; Significant p-values are shown in bold.

**Table 4. Investigation profile of the patients.**

| Variable | Total Mean ±SD | Recovered Mean ±SD | Died Mean ±SD | p-value |
|---|---|---|---|---|
| **Hemoglobin (g/dl)** | 12.2 ±0.2 | 12.2 ±0.3 | 12.1 ±0.5 | 0.800 |
| **RBC ($10^9$/mm$^3$)** | 4.44 ±0.2 | 4.3 ±0.2 | 4.7 ±0.2 | 0.394 |
| **WBC ($10^3$/mm$^3$)** | 9.9 ±0.5 | 9.4 ±0.6 | 10.9 ±0.6 | 0.154 |
| **Neutrophil (%)** | 73.1 ±1.4 | 71.6 ±1.6 | 76.9 ±2.5 | 0.091 |
| **Lymphocyte (%)** | 20.8 ±1.3 | 22.4 ±1.6 | 17.1 ±2.2 | 0.065 |
| **Platelet ($10^6$/mm$^3$)** | 237.6 ±12.8 | 240.9 ±16.5 | 229.5 ±19.1 | 0.694 |
| **ESR (mm)** | 22.8 ±1.7 | 22.7 ±2.2 | 23.4 ±2.6 | 0.853 |
| **RBS (mmol/l)** | 7.1 ±0.5 | 7.3 ±0.7 | 6.8 ±0.3 | 0.604 |
| **Serum calcium (mg/dl)** | 7.9 ±0.03 | 7.9 ±0.05 | 7.9 ±0.03 | 0.765 |
| **Serum creatinine (mg/dl)** | 1.2 ±0.1 | 1.2 ±0.1 | 1.2 ±0.1 | 0.948 |
| **Serum electrolyte (mmol/l)** | | | | |
| **Sodium** | 139.3 ±0.9 | 138.0 ±0.9 | 141.9 ±1.8 | **0.034** |
| **Potassium** | 4.4 ±0.7 | 3.7 ±0.1 | 5.6 ±1.8 | 0.178 |
| **Chloride** | 107.5 ±1.02 | 106.2 ±1.1 | 109.8 ±1.9 | 0.095 |

RBC: Red blood cell; WBC: White blood cell

p-value determined by independent samples t test; Significant p-values were shown in bold.

**Table 5. List of complications among patients (n = 61).**

| Variable | Total n (%) | Recovered n (%) | Died n (%) | p-value |
|---|---|---|---|---|
| **Any complication** | | | | |
| Present | 26 (42.6) | 13 (32.5) | 13 (61.9) | **0.027** |
| Absent | 35 (57.4) | 27 (67.5) | 8 (38.1) | |
| **Hypoxemia** | | | | |
| Present | 19 (31.1) | 6 (15.0) | 13 (61.9) | 0.164 |
| Absent | 42 (68.9) | 34 (85.0) | 8 (38.1) | |
| **Aspiration Pneumonia** | | | | |
| Present | 12 (19.6) | 5 (12.5) | 7 (33.3) | 0.05 |
| Absent | 57 (93.4) | 39 (97.5) | 18 (85.7) | |
| **Bedsore** | | | | |
| Present | 8 (13.1) | 7 (17.5) | 1 (4.8) | 0.161 |
| Absent | 53 (86.9) | 33 (82.5) | 20 (95.2) | |
| **Dysautonomia** | | | | |
| Present | 4 (6.5) | 2 (5.0) | 2 (9.5) | 0.498 |
| Absent | 57 (93.4) | 38 (95.0) | 19 (90.5) | |
| **DVT** | | | | |
| Present | 3 (4.9) | 1 (2.5) | 2 (9.5) | 0.228 |
| Absent | 58 (95.1) | 39 (97.5) | 19 (90.5) | |
| **UTI** | | | | |
| Present | 2 (3.3) | 2 (5.0) | 0 | 0.297 |
| Absent | 59 (96.7) | 38 (95.0) | 21 (100.0) | |
| **Sepsis** | | | | |
| Present | 1 (1.6) | 1 (2.5) | 0 | 0.465 |
| Absent | 60 (98.4) | 39 (97.5) | 21 (100.0) | |
| **Thrombophlebitis** | | | | |
| Present | 1 (1.6) | 0 | 1 (4.8) | 0.164 |
| Absent | 60 (98.4) | 40 (100.0) | 20 (95.2) | |
| **Wound infection** | | | | |
| Present | 1 (1.6) | 1 (2.5) | 0 | 0.465 |
| Absent | 60 (98.4) | 39 (97.5) | 21 (100.0) | |

p-value determined by Chi-square test; Significant p-values are shown in bold; DVT: Deep Vein Thrombosis; UTI: Urinary Tract Infection.

widespread in the environment, its eradication is impossible. On the other hand, life-long immunity against tetanus requires three booster doses of vaccine during the adolescent years [13]. Currently, only women of childbearing age are targeted through booster vaccination programs, which have substantially reduced maternal and neonatal tetanus in many countries. However, it remains a problem for adults in South Asia and Sub-Saharan Africa. The Global Disease Burden (GBD) studies suggested that, as of 2017, approximately 82% of all tetanus cases in the world were comprised of patients from these two regions, along with 77% of the total 38,000 tetanus deaths [3]. Hence, it was pertinent to study the factors associated with mortality among patients who are already infected. This study is one of the few attempts in Bangladesh to explore the factors associated with in-hospital mortality of adult tetanus patients.

We found that nearly 34.4% of patients died in the hospital. This is higher than two previous studies from Bangladesh, where authors reported 22.5% [14] and 28.6% [5] deaths. However, the observed death rates show regional variations. For instance, Tanon et al. reported a

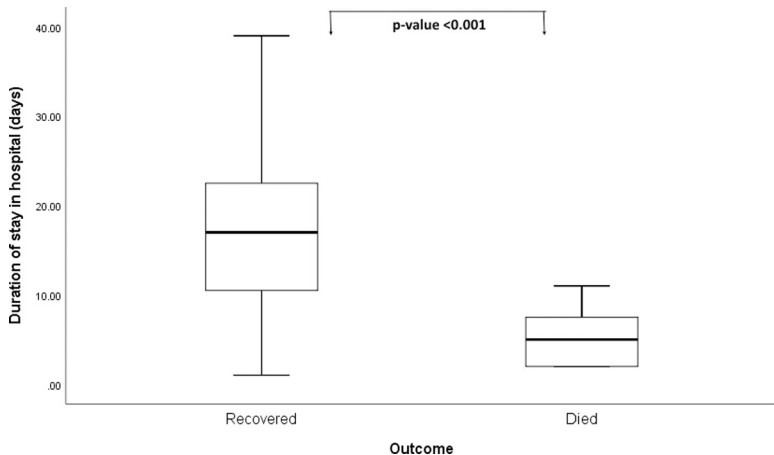

**Fig 2. Boxplots showing the duration of hospital stay among tetanus patients categorized by outcome**

death rate of 30% in the Ivory Coast [15], and Marulappa et al. found it to be 42.2% in India [16]. The case-fatality rates appear to be considerably lower where patients were provided ventilatory support. This can be seen in China [17] and Nepal [18], where only 5.9% and 7.5% of

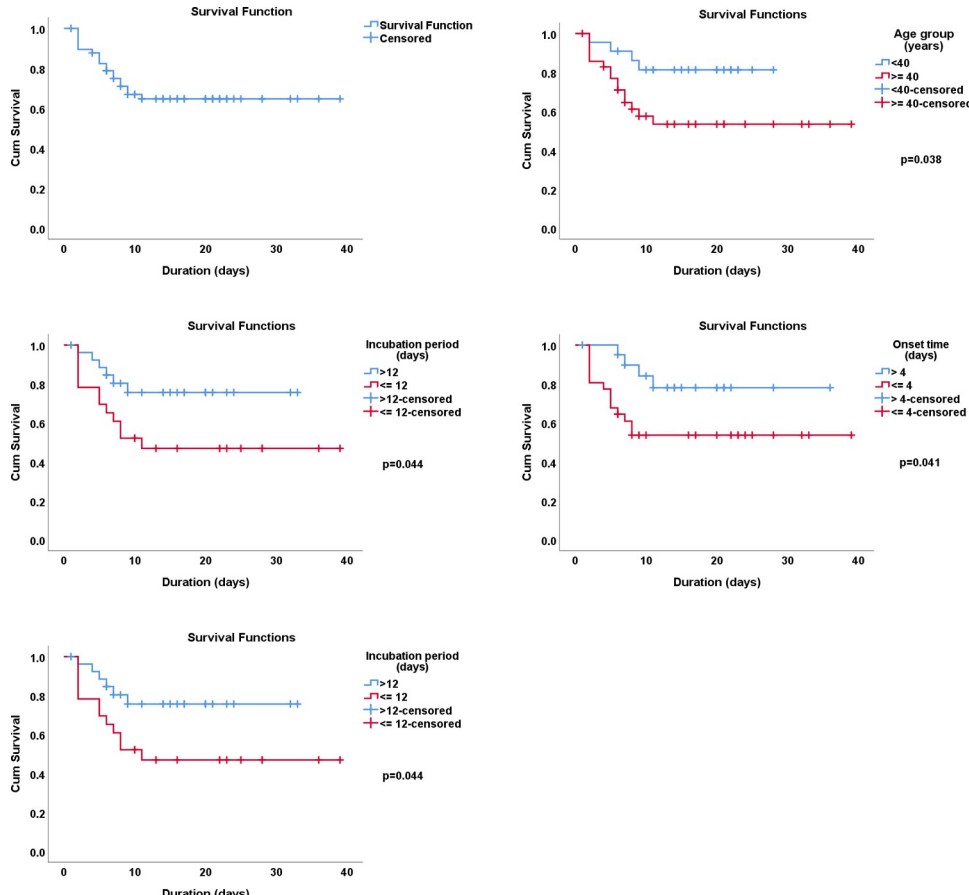

**Fig 3. Kaplan-Meier survival curves showing the probability of overall survival and survival across different groups of tetanus patients**

**Table 6. Univariate and multivariable Cox regression analysis for factors associated with in-hospital mortality among tetanus patients.**

| Factors | Reference Category | Crude HR (95%CI) | p-value | Adjusted HR (95%CI) | p-value |
|---|---|---|---|---|---|
| Age group (≥40 years) | <40 years | 2.98 (0.99–9.02) | 0.053 | 4.03 (1.07–15.17) | **0.039** |
| Incubation period (≤12 days) | >12 days | 2.57 (0.96–6.85) | 0.059 | 2.49 (0.90–6.90) | 0.078 |
| Onset time (≤4 days) | >4 days | 2.94 (0.97–8.97) | 0.058 | 3.33 (1.05–10.57) | **0.041** |
| Complication (Present) | Absent | 2.79 (1.06–7.34) | **0.038** | 2.09 (0.72–6.07) | 0.175 |

HR: Hazard ratio; Significant p-values are shown in boldface.

deaths were reported, respectively. However, despite giving mechanical ventilation, the death rate was 43.1% in Tanzania [19], a lower-middle-income country, which raises the importance of overall management facilities as well as other factors associated with deaths among tetanus patients.

We noted that patients who died had a significantly higher mean age than those who were alive, and patients with age ≥ 40 years had an increased chance of death in the hospital. This finding is supported by previous observations in Bangladesh [5], Tanzania [19], and India [16]. Similar to ours, Chalya et al. [19] noted that patients aged ≥40 years were significantly more likely to die when adjusted for other factors, such as incubation period, onset time, presence of complications, prior immunization, and tetanus severity.

The male population constituted the maximum number of participants in our study as well as other studies reporting on tetanus cases [7,14–19]. This probably reflects the outcome of immunization strategies for women of childbearing age in all countries. The WHO recommends three booster dosages of vaccine at ages: 12–23 months, 4–7 years, and 9–15 years for achieving life-long immunity for all [13]. Although many countries have programs for childhood immunization, expanding those to include booster dosages remains challenging. Hence, the vulnerable male population should be identified and vaccinated to protect them from tetanus.

One important finding of our study is that more than half of the patients were farmers and lived in rural areas. Similarly, many previous studies [16,19–22] observed a high proportion of farmers among tetanus patients. As *C. tetani* spores remain in the environment, people working in the fields, often with sharp cutting instruments, have a higher chance of being exposed to the bacteria due to accidental punctures or lacerated wounds. Farmers or agriculturists in low- and middle-income countries often work barefooted without adequate personal protective measures in the fields [23]. Therefore, they could be treated as a high-risk group to prioritize primary and booster immunization wherever appropriate.

We found that only 6.6% of patients could remember taking tetanus vaccine in the past and only one-quarter of patients took postexposure prophylaxis. Similar observations were reported by Lau et al. [21] and Tosun et al. [6]. Patients tend to forget the previous history of vaccination, and their attendants are even less likely to know about their vaccination history, which explains the reports. However, a low frequency of post-exposure prophylaxis indicates a lack of knowledge, awareness, and practice regarding tetanus vaccination after injury, particularly minor injuries, among the general population. Health authorities should work to address this issue where tetanus is not uncommon.

Although an onset time (duration between onset of symptoms to full-blown spasms) of 48 hours (2 days) and incubation of period of 7 days is used as a cutoff point of determining the increased risk of fatality in the majority of studies exploring prognostic factors of tetanus [2], we selected a higher cutoff point for both because the median onset time and incubation period were high among our study participants. Our analysis revealed that an onset time

of $\leq$ 4 days and an incubation period of $\leq$12 days significantly increased the chance of fatality of tetanus patients in the hospital. However, after adjustment for age and the presence of complications, only onset time ($\leq$4 days) remained a significant determinant of in-hospital mortality. Previously, Amare et al. [11] reported a univariate association of <3 days onset time with mortality, which became statistically nonsignificant after multivariate logistic adjustments. Similarly, Krishnan et al. [12] and Chalya et al. [19] didn't find any association of onset time ($\leq$48 hours) with the death of tetanus patients. The incubation period of <7 days was found to be associated with a significantly increased risk of mortality in several previous reports [16,19]. In contrast, some other reports did not find such an association [7,11]. However, very few studies have considered a higher incubation period cutoff point for analysis, making it difficult to comment on the findings. Additionally, differences in sample size and consideration of other factors during multivariate adjustment as well as the type of analysis used, often influence the outcome. Hence, the findings should be read in the context of a particular study.

The clinical presentation of tetanus shows variability in previous studies. The most common symptoms among our patients included lockjaw (trismus), dysphagia, and spasticity like previous observations in Bangladesh [5,14]. We did not find any noticeable differences in clinical features between alive and dead tetanus patients. Neither did we find any remarkable differences in routine investigations of dead and alive patients. Serum sodium levels were slightly but statistically significantly higher in deceased patients than that of recovered ones. However, both groups had their values within the normal limit. Therefore, the difference discovered could be a random finding that needs to be evaluated through further large sample studies.

Unlike Chalya et al. [19], we did not find an increased risk of mortality in patients with severe tetanus. But our finding is concordant with that of Amare et al. [11]. However, it might depend on the scales used for severity grading among patients. The Ablett classification scheme used in our study is dependent on the presence of spasms and autonomic disturbance, which evolve throughout the course of the disease [2]. Hence it is considered to be less suitable for prognostic stratification. Nevertheless, similar to other studies [22], we noted that nearly 50% of patients presented with severe and very severe disease.

In the present study, only 8.2% of participants had one or more comorbidities from diabetes, hypertension, COPD, and ischemic heart disease. However, it was not associated with a higher risk of mortality. Rather, the development of complication(s) was associated with a significantly higher risk of death in the hospital, which is concordant with findings from Marulappa et al. [16], Tanon et al. [15], Krishnan et al. [12], and Bankole et al. [20]. Hypoxemia and aspiration pneumonia were frequent complications that we encountered in our patients. Hypoxemia is the consequence of laryngeal spasms, respiratory distress, and respiratory failure that might occur in tetanus patients [12]. Unlike Amare et al. [11], who reported the occurrence of dysautonomia in 91.7% of deceased tetanus patients, we found the problem in only 9.5% of patients who died. This could be explained by the lack of cardiac monitoring facilities and intensive care units in our centers, the presence of which would have allowed close monitoring and precise detection of autonomic imbalance in these patients. Moreover, life-saving ventilatory supports for patients were not possible either because of the lack of facilities.

On multivariable regression analysis, after adjusting for incubation period and presence of complications, age $\geq$ 40 years and onset time of $\leq$4 days were found to be significant predictors of in-hospital deaths of patients. Aging is associated with a reduction in antibody levels in individuals previously immunized against tetanus [24,25]. Moreover, aging itself shows a decline of the immune system [26], and a rise in the number of chronic conditions [27], leading to a lower healing capacity. All of these factors might combinedly influence the outcome of tetanus in older patients. A short onset time implies a quick evolution of the disease after initiation of symptoms leading to increased severity [2]. Hence, aggressive management strategies

are required for patients ≥40 years of age and short onset time. Moreover, assisted ventilation facilities are crucial for critically ill tetanus patients. The high proportion of death observed in this study could be attributed to the lack of intensive care facilities in these centers.

## Limitations

Our study had several limitations. The sample size was small. The impact of assisted ventilation on mortality could not be evaluated due to a lack of facilities. Further follow-up of patients after discharge from the hospital was also not possible. However, this was one of the few attempts to assess the clinical-epidemiological profile and factors associated with in-hospital mortality among tetanus patients in Bangladesh. Our findings would undoubtedly spark interest in further studies and inform policymakers regarding the limitations in the management of tetanus that need to be addressed.

## Recommendations

Considering the findings of this study, we have the following recommendations-

1. Adult tetanus patients with a higher age should be given special care during management.

2. A shorter onset time of tetanus must warrant careful assessment of disease severity for meticulous treatment.

3. A booster vaccination program should be started prioritizing vulnerable male population to prevent tetanus incidence in the country.

4. Further large-scale countrywide studies on tetanus patients should be carried out to explore management strategies and reduce case-fatality among tetanus patients.

## Conclusion

Although expanded programs on immunization and maternal and neonatal tetanus elimination programs have been successful in the considerable reduction of tetanus in neonates, children, women of childbearing age, and men at their early adulthood, older adult men are still vulnerable to tetanus in Bangladesh. As most of the cases come from rural areas where farming and manual work are the principal modes of earning, they could be considered a priority group for a booster vaccination program. On the other hand, tetanus patients with higher age and shorter onset time needs special care during management as they have a higher risk of death.

## Supporting information

**S1 Table. Questionnaire.**
(DOCX)

**S2 Table. Ablett classification of tetanus severity.**
(DOCX)

## Acknowledgments

Authors acknowledge the contribution of the patients and their guardians for their kind consent to participate in the study.

## Author Contributions

**Conceptualization:** Md. Abdullah Saeed Khan, Mohammad Hayatun Nabi, Nadira Sultana Kakoly.

**Data curation:** Md. Abdullah Saeed Khan.

**Formal analysis:** Md. Abdullah Saeed Khan.

**Investigation:** Md. Abdullah Saeed Khan, Md. Utba Rashid, Soumik Kha Sagar, Sanzida Khan.

**Methodology:** Md. Abdullah Saeed Khan, Mohammad Jahid Hasan, Susmita Zaman, Sultan Mahamud Sumon, Ariful Basher, Mohammad Delwer Hossain Hawlader, Mohammad Hayatun Nabi, Nadira Sultana Kakoly.

**Project administration:** Md. Abdullah Saeed Khan, Md. Utba Rashid, Soumik Kha Sagar, Sanzida Khan.

**Resources:** Md. Abdullah Saeed Khan, Mohammad Jahid Hasan, Soumik Kha Sagar, Sanzida Khan, Susmita Zaman.

**Software:** Md. Abdullah Saeed Khan, Mohammad Jahid Hasan, Susmita Zaman.

**Supervision:** Md. Abdullah Saeed Khan, Sultan Mahamud Sumon, Ariful Basher, Mohammad Delwer Hossain Hawlader, Mohammad Hayatun Nabi, Nadira Sultana Kakoly.

**Validation:** Md. Utba Rashid.

**Visualization:** Md. Abdullah Saeed Khan.

**Writing – original draft:** Md. Abdullah Saeed Khan.

**Writing – review & editing:** Md. Abdullah Saeed Khan, Mohammad Jahid Hasan, Md. Utba Rashid, Soumik Kha Sagar, Sanzida Khan, Susmita Zaman, Sultan Mahamud Sumon, Ariful Basher, Mohammad Delwer Hossain Hawlader, Mohammad Hayatun Nabi, Nadira Sultana Kakoly.

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
