## [Decision Letter · Decision Letter 0]

13 Jan 2022

Dear Dr. Khan,

Thank you very much for submitting your manuscript "Factors Associated with In-Hospital Mortality of Adult Tetanus Patients– A Multicenter Study from Bangladesh" for consideration at PLOS Neglected Tropical Diseases. As with all papers reviewed by the journal, your manuscript was reviewed by members of the editorial board and by several independent reviewers. The reviewers appreciated the attention to an important topic. Based on the reviews, we are likely to accept this manuscript for publication, providing that you modify the manuscript according to the review recommendations. 

Sincerely,

Joseph M. Vinetz

Deputy Editor

Joseph Vinetz

Deputy Editor

Reviewer's Responses to Questions

**Key Review Criteria Required for Acceptance?**

**Methods**

-Are the objectives of the study clearly articulated with a clear testable hypothesis stated?

-Is the study design appropriate to address the stated objectives?

-Is the population clearly described and appropriate for the hypothesis being tested?

-Is the sample size sufficient to ensure adequate power to address the hypothesis being tested?

-Were correct statistical analysis used to support conclusions?

-Are there concerns about ethical or regulatory requirements being met?

Reviewer #1: (No Response)

Reviewer #2: The objectives of the study are clearly articulated. The study did not require testing any hypothesis.

The study design was appropriate to address the objectives stated.

The population studied was clearly described and appropriate for the objectives

The sample size is good for the study

The Statistical analysis used were correct to support the conclusions

The ethical requirement will require some modification as the authors claimed that patients with tetanus gave written informed consent. This calls for questioning as a patient with spasm may be unable to give written consent, however the patient may be able to give verbal consent with their caregiver going ahead to sign any document that needs to be signed.

Reviewer #3: Objectives are clear; study design is relatively appropriate; description of the population needs some clarity- example who are excluded with reasons to be mentioned; No concern about ethical aspects.

Who evaluated the patients in the two hospitals and how uniformity in assessment was maintained?

**Results**

-Does the analysis presented match the analysis plan?

-Are the results clearly and completely presented?

-Are the figures (Tables, Images) of sufficient quality for clarity?

Reviewer #1: (No Response)

Reviewer #2: The analysis presented matched the analysis plan

The results were clearly presented except for the Kaplan Meier Survival Analysis that needs to be reviewed as presently presented by the authors. Table 1 requires some corrections which have been highlighted.

The images and figures are clear to a good extent except for figure 3 which is not so clear.

Reviewer #3: Table 1: need correction- widow; income per month

Lines 195-196- not clear

Definition of post exposure prophylaxis?

What is meant by mode of injury and sub-types?

Definition of 'localized' and generalized' tetanus may be given

Time of onset- needs clarification

Definition of 'severity score' and 'trismus severiity score' may be given in the method.

Any photo of 'trismus' if available with consent may be provided

**Conclusions**

-Are the conclusions supported by the data presented?

-Are the limitations of analysis clearly described?

-Do the authors discuss how these data can be helpful to advance our understanding of the topic under study?

-Is public health relevance addressed?

Reviewer #1: (No Response)

Reviewer #2: The conclusions are supported by the data presented

The limitations of the study were fairly well described

The authors did discuss the relevance of the study viz-a-viz tetanus elimination.

Tetanus is a disease of public Health relevance and this was highlighted in the manuscript.

Reviewer #3: Limitations of the study: facilities of intensive care assisted respiration in hospitals managing the cases of tetanus may be made available in Bangladesh

Some part of data needs clarity

Public relevance addressed well

**Editorial and Data Presentation Modifications?**

Reviewer #1: (No Response)

Reviewer #2: The authors have tried. However, the manuscript has lots of errors of syntax and semantics which requires a lot of work.

Reviewer #3: Minor Revision

**Summary and General Comments**

Reviewer #1: (No Response)

Reviewer #2: Thanks for asking me to review this manuscript. The topic is of public health relevance. It highlighted the need to focus on adult farmers who are high risk as far as tetanus is concerned. Some concerns have been highlighted as sticky notes in the manuscript and some suggestions made.

Reviewer #3: (No Response)

PLOS authors have the option to publish the peer review history of their article (what does this mean?). If published, this will include your full peer review and any attached files.

Reviewer #1: No

Reviewer #2: No

Reviewer #3: No

Figure Files:

Data Requirements:

Reproducibility:

References

---

## [Editor Report · Decision Letter 1]

5 Feb 2022

Dear Dr. Khan,

We are pleased to inform you that your manuscript 'Factors Associated with In-Hospital Mortality of Adult Tetanus Patients– A Multicenter Study from Bangladesh' has been provisionally accepted for publication in PLOS Neglected Tropical Diseases.

Best regards,

Joseph M. Vinetz

Deputy Editor

Joseph Vinetz

Deputy Editor

---

## [Editor Report · Acceptance letter]

14 Feb 2022

Dear Dr. Khan,

We are delighted to inform you that your manuscript, "Factors Associated with In-Hospital Mortality of Adult Tetanus Patients– A Multicenter Study from Bangladesh," has been formally accepted for publication in PLOS Neglected Tropical Diseases.

Best regards,

Shaden Kamhawi

co-Editor-in-Chief

Paul Brindley

co-Editor-in-Chief
